# Neck Stabilization Exercises Enhance Respiratory Function after Stroke: Respiratory Function Index Change Trajectory Analyzed Using a Hierarchical Linear Model

**DOI:** 10.3390/medicina57121312

**Published:** 2021-11-30

**Authors:** So-Hyun Kim, Sung-Hyoun Cho

**Affiliations:** 1Department of Medical Sciences, Graduate School, Nambu University, 23, Cheomdan Jungang-ro, Gwangsan-gu, Gwangju 62271, Korea; kimso3716@naver.com; 2Department of Physical Therapy, Nambu University, 23, Cheomdan Jungang-ro, Gwangsan-gu, Gwangju 62271, Korea

**Keywords:** deep neck muscles, hierarchical linear model, neck stabilization, paralysis, respiratory function, stroke

## Abstract

*Background and Objectives*: This study aimed to assess the effect of neck stabilization exercise on respiratory function in stroke patients through longitudinal observation and determine whether there is a difference in its effect based on the side of paralysis in the patients. It is difficult to observe the amount of change observed in individuals and groups as most intergroup comparison studies only use mean values. To address these shortcomings, this study adopted a hierarchical linear model (HLM) in our trajectory analysis. *Materials and Methods*: We conducted neck stabilization training three times a week for four weeks in a single group of 21 stroke patients. To evaluate respiratory function, their forced vital capacity (FVC), forced expiratory volume in the first second (FEV1), forced expiration ratio (FEV1/FVC), and peak cough flow (PCF) were measured. Data analysis was performed using HLM 8.0. *Results*: A significant increase was found in the respiratory function after neck stabilization training (*p* < 0.05). While neck stabilization training overall was longitudinally effective, the growth rate of respiratory function in left-sided paralytic patients was less than the whole group value. Conversely, the growth rate of respiratory function in right-sided paralytic patients was greater than the whole group value. *Conclusions*: This study demonstrated that neck stabilization training is longitudinally effective in improving respiratory function in stroke patients. Additionally, the growth rate of respiratory function was greater in patients with right side paralysis than in patients with left side paralysis.

## 1. Introduction

Stroke is the leading cause of death worldwide and is a representative disability. More than 70% of stroke patients develop chronic disability due to general muscle weakness; this affects even the respiratory muscles, such as the diaphragm and intercostal muscles, with invasion of these principal respiratory muscles typically seen on the paralyzed side of stroke patients [1,2]. The diaphragm is the inspiratory muscle responsible for most of the total ventilation of an individual in the sitting and lying positions, and its paralysis causes dysfunction in 51.7% of stroke patients as it leads to a marked decrease in respiratory function [3]. At the same time, a secondary problem caused by muscle paralysis is that the lungs and chest cage fail to achieve sufficient inflation, which leads to asymmetric respiration due to the following: decreased lung compliance, abnormal chest expansion, increased sensitivity to carbon dioxide, and decreased voluntary respiration [4,5]. It was determined that muscle paralysis in stroke patients leads to weakened respiratory function, including forced vital capacity (FVC), forced expiratory volume in the first second (FEV1), forced expiration ratio (FEV1/FVC), and peak cough flow (PCF), which is accompanied by impaired daily living activities, social restrictions, and respiratory complications [6,7]. Pulmonary function tests are mainly used to diagnose a decrease in respiratory volume by measuring respiratory function [8]. FVC refers to the amount of air at maximum inhalation and maximum exhalation efforts. FEV1 refers to the maximum amount exhaled over 1 s at the maximal inspiratory level. The FEV1/FVC ratio can estimate airway obstruction. PCF refers to the volume of air forced out of the lungs in one rapid exhalation. Measurements from these lung function tests are indicators of either adequate ventilation or airflow obstruction [8].

To assist the paralysis of the principal respiratory muscle and compensate for the decreased respiratory function of stroke patients, overuse of the respiratory accessory muscles, such as the sternocleidomastoid and scalenus muscles, may occur and may move the head forward [9]. This postural change affects the anteroposterior diameter of the chest cage as well as the volume of the lungs, thereby affecting the respiratory system [10]. Based on this approach, a previous study tested maximal voluntary ventilation [11] and FVC [12] improvement in stroke patients after neck stabilization training. In addition, the limitation of the neck muscles and joint range of motion causes changes in the movement of the chest cage based on the movement of the neck, and the inappropriate use of the relevant muscles of the principal respiratory muscle causes a decrease in respiratory function [13]. In a previous study, the correlation between the movement of the diaphragm, the main respiratory muscle, and respiratory function in stroke patients depended on the side (left or right) of the paralysis [14]. In other words, left-sided paralysis has a different impact on respiratory function than right-sided paralysis. In particular, an imbalance in neck stabilization leads to excessive use and shortening of the shallow accessory muscles of respiration and functional obstruction of deep neck respiration, which leads to respiratory failure [15]. Neck stabilization abnormalities are also accompanied by pain due to postural misalignment, resulting in continued physical stress [16]. Considering this, the functional recovery and departure of stroke patients from the rehabilitation treatment process is inevitably slow. Therefore, respiratory function and other therapeutic effects can potentially be maximized through neck stabilization training. Moreover, respiratory function between the paralyzed sides should be compared. Any differences between paralyzed sides may provide guidance for future neck stabilization training.

Although it is known that respiratory function deterioration in stroke patients is common in clinical practice, there is one aspect that is commonly overlooked in relation to breathing because changes in respiratory function in stroke patients do not cause obvious respiratory problems. However, from a social point of view, recent and prolonged outbreaks of coronavirus disease 2019 (COVID-19) are associated with a wide range of neurological complications, including stroke, as well as deterioration of lung function, raising the risk of deterioration [17]. Since the onset of stroke and deterioration of lung function can be fatal and unexpected, research to improve the respiratory function of stroke patients is very important. In various studies focusing on improving respiratory function in stroke patients, trunk stabilization training for activation of the principal respiratory muscles of stroke patients [18,19] as well as joint range of motion training in the spine and cervical region have already been conducted [20]. However, there are insufficient studies focused on restoring respiratory function of stroke patients by improving the stabilization of the neck muscles [11,12]. In particular, in terms of stroke patterns, the severity of different symptoms varies from acute to chronic with observable differences between their presentation in individuals; however, there are no studies observing changes in terms of respiratory function based on the unique neck stabilization of each individual.

Therefore, unlike previous studies, this study aimed to observe the effects of applying neck stabilization training to stroke patients, checking the degree of longitudinal changes in respiratory function, at both the group level and individual level. Furthermore, factoring in the side of paralysis as an independent variable, the amount of change and difference between each group and individual, as well as the effect of neck stabilization training on the improvement of respiratory function, were investigated.

## 2. Methods

### 2.1. Participants

This study was conducted with 21 stroke patients hospitalized at the C Rehabilitation Hospital in Gwangju, Korea. The study design was approved by the Institutional Review Board (IRB) of Nambu University, Gwangju, Republic of Korea (NBU-IRB-1041478-201503-HR-006) and was granted clinical trial registration (KCT0001555); this study was conducted in accordance with the Declaration of Helsinki. All participants understood the purpose of the study and signed written informed consent forms. The specific selection criteria for participants were as follows: (1) at least six months have passed since the onset of stroke, (2) no history of respiratory disease or injury, (3) no lung disease upon radiological examination and physical examination of the chest, and (4) without cognitive dysfunction expected to make the individual uncooperative with the experiment due to severe aphasia or dementia (Korean version of Mini-Mental Status Examination; K-MMSE score of 24 or higher).

### 2.2. Training Method

Neck stabilization training was performed three times a week for four weeks, and the training method was sequentially performed according to a previous study [11]. Neck stabilization training was performed with a low load to strengthen the longus capitis and longus colli, which are deep muscles of the upper cervical vertebra [21,22]. Relaxation of the oblique and anterior rib muscles, which are shallow respiratory auxiliary muscles, was maintained, and the cranial neck was also flexed and maintained. A pressure biofeedback device (Chattanooga, Hixson, TN, USA) was placed in the upper cervical spine (below the occiput) while lying down with the air pocket set to 20 mmHg in order to obtain visual feedback from the dial. The device was used for activation and strengthening of the neck muscles by providing a constant contractile force specifically to the deep neck muscles. An experienced researcher demonstrated an approach that prevents the use of auxiliary respiratory muscles before proceeding with the neck stabilization training. For correct neck stabilization training, the researcher contacted the subject’s sternocleidomastoid muscle and scalene muscle to confirm that no contractions occurred. The pressure was gradually increased to 30 mmHg by increments of 2 mmHg. While asking the patient to retract his or her chin, the contraction was held for 10 s and repeated 10 times. A rest period of 3–5 s was given in between each contraction (Table 1, Figure 1A,B).

### 2.3. Measurement

Respiratory function and coughing ability were measured 12 times, or three times a week for four weeks, immediately after neck stabilization training. Respiratory function evaluation was performed using the respiratory function calculator of a spirometer (MicroLab 3300 Spirometer MK4, Micro Medical Ltd., Chatham, UK); forced vital capacity (FVC), forced expiratory volume in the first second (FEV1), and forced vital capacity ratio (FEV1/FVC ratio) were measured. Using the electric angle control device of the bed, the upper body was erected to approximately 60°, and the patient was instructed to breathe only through the mouth while using a nose plug. FVC refers to the amount of air exhaled as quickly as possible after maximum inhalation of the patient, and FEV1 refers to the amount of air exhaled strongly for one second after maximum inhalation. Each test was performed three times, and the average value was recorded. To evaluate coughing ability, peak cough flow (PCF) was measured using a peak flow meter (Mini-wright AFS Low range peak flow meter, Cardinal Health 232 Ltd., Basingstoke, UK). PCF evaluation was performed in the same position as the previous evaluations, and the patient was asked to cough as strongly as possible after inhaling as much air as possible. Each test was performed three times, and the average value was recorded.

### 2.4. Data Analysis

Statistical analyses were performed using SPSS software for Windows, version 25.0 (SPSS Inc., Chicago, IL, USA) and HLM version 8.0 (Scientific Software International Inc., Skokie, IL, USA). The Shapiro–Wilk test was used to verify the normality of the sample data. The two-sample *t*-test was used to compare the left- and right-paralyzed groups. The detailed analysis method included the following: frequency, percentage, mean, and standard deviation were used to exhibit the descriptive statistics of the study subjects’ general characteristics and the study variables. Statistical significance was considered to be a *p*-value of less than 0.05 (*p* < 0.05) using a two-tailed test. For the experimental effect analysis, a multi-layered growth model was investigated to check how the study variables at the paraplegic side level and the individual level affect respiratory function index linear growth over time.

Level-1 of the multi-layered growth model utilized repeated observations. To test whether individual respiratory function significantly changed over time, the respiratory function was observed over 12 sessions. Level-2 accounted for individual differences. If the value of level-1 was significant, the location of the paralysis was analyzed as an independent variable. If the random effect was statistically significant, individual differences in initial value (*ψ_0j_*) and rate of change were explained by adding an independent variable (*PARETIC_j_*). In other words, level-2 identified the differences in the initial value (*ψ_0j_*) and the rate of change between individuals.

#### 2.4.1. Null Model

The basic model estimates the longitudinal change of the dependent variable, respiratory function, by investing time in level-1. At the same time, it checks whether the difference between the initial value (*ψ_0j_*) and the rate of change between individuals is significant. This model determines whether future research models will be meaningful. It achieves that by inputting independent variables (*PARETIC_j_*) to level-2 of the basic model in the future research model [23]. The composition of the basic model is presented in Table 2.

*FVC_mj_* is a measure of individual *𝑚* in group *j* and corresponds to the dependent variable of the model. *TIME_mj_* is an independent variable for predicting or explaining the dependent variable as a characteristic variable of the individual *𝑚* in group *j*, and *ψ_0j_* is the intercept regression coefficient for the influence of the mean with level-1 coefficients. *ψ_1j_* is the slope coefficient of the level-1 coefficients. *𝑒_mj_* is a level-1 random effect, and with respect to the measured value of individual *𝑚* in group *j*, it is a residual value that is not explained by an independent variable—a random error. The level-2 model is used to predict or explain the level-1 coefficients *ψ_0j_* and *ψ_1j_*.

#### 2.4.2. Research Model

Based on the significant results of the null model, the need to input an independent variable in level-2 was proved, and the side of paralysis of stroke patients was input as a variable; in the research model, the growth model was applied and analyzed to examine the effects of side of paralysis on respiratory function and cough flow (Table 3).

*PARETIC_j_* is a level-2 independent variable and is a characteristic variable of group *j*. *γ_00_*, *γ_01_*, *γ_10_*, and *γ_11_* are level-2 coefficients and are fixed effects parameters. *γ_00_* and *γ_10_* are intercepts of the level-2 model. *γ_01_* is the effect of the level-2 independent variable (*PARETIC_j_*), and *γ_11_* is the interaction effect between the level-1 independent variable (*TIME_mj_*) and the level-2 independent variable (*PARETIC_j_*). *𝑒_mj_*, *𝑢_0j_*, and *𝑢_1j_* are the random effect parameters. *𝑢_0j_* and *𝑢_1j_* are level-2 random effects and are residuals for each group that is not explained by the group-level independent variable (*PARETIC_j_*).

## 3. Results

### 3.1. General Characteristic of Research Subjects

Table 4 shows the distribution of the selected experimental group.

When looking at the side of paralysis of the research participants, eight patients (38.1%) were paralyzed on the right side and 13 (61.9%) were paralyzed on the left side; the proportion of patients with paralysis on the left side was higher. There were four males (19.1%) and 17 females (80.9%). The average age of all patients was 76.00 ± 10.19 years old, and the average height was 159.86 ± 6.93 cm. Additionally, the average weight of all patients was 53.56 ± 9.27 kg with a mean BMI of 20.96 ± 3.45 kg/m^2^. The average duration of the first stroke was 32.29 ± 18.00 months. There was no significant difference observed in terms of physical characteristics based on side of paralysis (*p* > 0.05).

### 3.2. Variables Used in the Analysis

Table 5 presents the descriptive statistics of the variables used in the analysis including the average value after 12 training sessions. The table showed an overall increase in lung function.

### 3.3. Null Model

The change in the linear growth rate (*ψ_1j_*) of the respiratory function exhibited by the whole group is shown in Table 6. A *p* < 0.05 was considered to be statistically significant. The threshold was set to 1.96, and if the absolute value of the t-statistic was greater than this threshold, the null hypothesis H0 was rejected.

The linear growth rate (*ψ_1j_*) of the FVC score of all stroke patients during the measurement period was 33.345, which was statistically significant. The average FVC of the whole group improved by 33.345 mL (Figure 2A). The linear growth rate (*ψ_1j_*) of the FEV1 score of all stroke patients during the measurement period was 31.086, which was statistically significant. The average FEV1 of the whole group improved by 31.086 mL (Figure 2B). The linear growth rate (*ψ_1j_*) of the FEV1/FVC ratio score of all stroke patients during the measurement period was 0.269, which was statistically significant. The FEV1/FVC ratio of the whole group improved by 0.269% on average (Figure 2C). The linear growth rate (*ψ_1j_*) of the PCF score of all stroke patients during the measurement period was 9.043, which was statistically significant. The average PCF of the entire group improved by 9.043 L/min (Figure 2D).

### 3.4. Research Model

The changes in respiratory function in the two groups are shown in Table 7.

There was a significant difference seen in the initial FVC scores of the left- and right-sided paralysis groups. The growth rate (*ψ_1j_*) of the former (−8.696) is smaller than the whole group value (*p* > 0.05), and the growth rate (*ψ_1j_*) of the latter (38.728) is bigger than the whole group value (33.345) (*p* < 0.01). The right-sided paralysis group showed a difference in the initial FVC, and it can be seen that the FVC improved by an average of 38.728 mL over time after performing neck stabilization exercises (Figure 2A).

There was a significant difference seen in the initial FEV1 scores of the left- and right-sided paralysis groups. The growth rate (*ψ_1j_*) of the former (−7.176) is smaller than the whole group value (*p* > 0.05), and the growth rate (*ψ_1j_*) of the latter (35.529) is bigger than the whole group value (31.086) (*p* < 0.01). The right-sided paralysis group showed a difference in the initial FEV1, and it can be seen that the FEV1 improved by an average of 35.529 mL over time after performing neck stabilization exercises (Figure 2B).

There was a significant difference seen in the initial values of FEV1/FVC of the left- and right-sided paralysis groups. The growth rate (*ψ_1j_*) of the former (0.296) is bigger than the whole group value (*p* > 0.05), and the growth rate (*ψ_1j_*) of the latter (0.086) is smaller than the whole group value (0.269) (*p* > 0.05) (Figure 2C).

There was a significant difference seen in the initial values of PCF scores of the left- and right-sided paralysis groups. The growth rate (*ψ_1j_*) of the former (1.063) is smaller than the whole group value (*p* > 0.05), and the growth rate (*ψ_1j_*) of the latter (9.701) is bigger than the whole group value (9.043) (*p* < 0.01). The right-sided paralysis group showed a difference in the initial PCF, and it can be seen that the PCF improved by an average of 9.701 mL over time after performing neck stabilization exercises (Figure 2D).

## 4. Discussion

In this study, FVC, FEV1, FEV1/FVC ratio, and PCF were used to investigate the effect of neck stabilization exercise on group and individual respiratory function growth trajectory, also factoring in the side of paralysis of stroke patients.

Normal FVC values are 3600 mL for men and 2500 mL for women, and FEV1 are 2500 mL for men and 1800 mL for women [24]. Normal cough flow values are between 360 and 400 L/min [25]. After the first exercise, the right paralytic patients had an average FVC value of 1388.75 mL and an average FEV1 value of 1105.00 mL. The left paralytic patients had an average FVC value of 1620.77 mL and an average FEV1 value of 1350.77 mL. The average PCF values were 115.00 L/min for the right paralytic patients and 112.69 L/min for the left the paralytic patients. Overall, the lung function of the stroke patients in this study was decreased. After the 12th exercise, the right paraplegic patients had an average FVC value 1783.75 mL and an average FEV1 value of 1522.50 mL. The left paraplegic patients had an average FVC value of 1884.62 mL and an average FEV1 value of 1692.31 mL. Moreover, the average PCF values were 211.25 L/min for the right paralytic patients and 193.85 L/min for the left paralytic patients. In this study, the lung function of the stroke patients increased after multiple training sessions.

As such, in previous studies, most of the analyses conducted verified differences in the mean values of groups using a single-group, pre-post design method, or a pre-post control group design. Since these studies only show differences or effectiveness at each time point based on average values, it is not possible to accurately derive the individual growth trajectory of respiratory function at each stage. Essentially, these studies did not measure the amount of individual temporal change. In other words, it is impossible to determine whether each individual in the group had an effect on the basic value of the state and whether there was a difference between groups or individuals in terms of the growth rate they had exhibited. The hierarchical linear model (HLM), which was proposed as an appropriate model for identifying individual changes by addressing the above problems, is a statistical method for analyzing data with a hierarchical structure in which variables measured at a specific level are inherent at a higher level [26]. Analysis by HLM provides clearer data for related studies because it allows for distinguishing variance between group-level and individual-level variables within a multi-layered model [27].

The results of this study showed that the linear growth rate of all stroke patients was statistically significant during the measurement period. However, in the additional research model, the growth rate of the left-sided paralysis is smaller than the whole group value in FVC, FEV1, and PCF. The growth rate of the right-sided paralysis is bigger than the whole group value in FVC, FEV1, and PCF.

Joint range of motion exercise [20] and stabilization training [12] of the neck performed in comparison with the experimental group and control group for stroke patients showed results consistent with this study, exhibiting improvement of respiratory function. However, previous studies could not predict that neck stabilization training would be more effective in terms of improving respiratory function for patients with right-sided paralysis. To the best of our knowledge, this is the first longitudinal study comparing left- and right-sided paralysis of stroke patients in terms of their relation to neck stabilization, so it was speculated primarily based on the relationship of the principal respiratory muscles and the expansion of the chest cage.

According to a previous study, it can be seen that the decrease in motion amplitude of the left diaphragm is higher than that of the right side, even without considering the side of paralysis [28]. In addition, after unilateral injury, the paralyzed diaphragm of a stroke patient is passively attracted by the ipsilateral thoracic and non-paralyzed aspect of the diaphragm. To compensate for diaphragmatic paralysis, the respiratory activity and driving force of the thoracic cage and abdominal muscles increase, resulting in paradoxical thoracoabdominal respiration [29]. Expansion of the rib cage results from the cooperation of the accessory respiratory muscles, the sternocleidomastoid muscle, during inspiration [30].

This study attempted to achieve neck stabilization by limiting the use of accessory respiratory muscles as much as possible and inducing deep cervical flexor (DCF) muscles. Paralysis of the diaphragm is continuous, and the continuation of respiratory function cannot be limited only by training for neck stabilization. To compensate for diaphragmatic paralysis, left thoracic expansion is performed on the ipsilateral left oblique muscle of the neck, and even if the respiratory function is improved by securing the appropriate muscle length, we believe that it will be difficult for patients with left-sided paralysis to maintain the continuity of the respiratory function effect over time due to the compensatory action of the oblique occipital muscle.

Moreover, for dysphagia requiring comprehensive neck muscle intervention in stroke patients, shallow neck muscles such as the oblique neck muscles are mobilized first at the beginning of head bending training, and the activity of the oblique neck muscles decreases as deep neck muscle strengthening training progresses [31]. Although this study did not investigate the subject’s dysphagia, swallowing is related to the movement of the diaphragm, the principal respiratory muscle, as it shares several anatomical structures and muscles with respiratory function [32,33]. Dysphagia tends to appear longer in patients with left-sided paralysis [34]. In a recent study comparing dysphagia based on side of paralysis of stroke patients, it was reported that the endurance of the deep neck flexor muscle was superior in patients with right-sided paralysis compared to those with left-sided paralysis [35]. Parallel to this, the results of this study also showed that the respiratory function of patients with right-sided paralysis was already high at the initial value and also increased over time. Paradoxically, since it is difficult to maintain the endurance of the deep neck flexor muscles in patients with left-sided paralysis, it is judged that although the respiratory function was longitudinally effective, the growth rate of left-sided paralysis is smaller than the whole group value in FVC, FEV1, and PCF.

However, a major limitation of this study is that it could not be generalized to the improvement of respiratory function in all stroke patients due to the small study population. Another limitation is the inconsistent spacing between sessions as there are 3 trainings/week for 4 weeks. In addition, it is necessary to subdivide the model based on the characteristics of patients with stroke. This is because there are some aspects that could actually be overlooked depending on the number of participants in relation to the type of stroke and characteristics of stroke patients. Therefore, in future investigations, it will be necessary to conduct multifaceted research on more stroke patients when determining and visualizing the growth rate of respiratory function.

## 5. Conclusions

The results of this study revealed that neck stabilization training is effective for improving respiratory function in stroke patients as time increases. After examining individual growth rates and the effect of neck stabilization training on the entire stroke patient group, we noted that patients with right side paralysis exhibited an increase in the growth rate of respiratory function when compared to patients with left side paralysis, and when compared to the whole group. These results are significant in comparison to previous studies because the hierarchical linear model (HLM) was applied to identify the growth trajectories of groups and individuals by stage. It is expected that more specific and systematic analysis methods, such as HLM, will be utilized in various studies in the field of physical therapy.

## Figures and Tables

**Figure 1 medicina-57-01312-f001:**
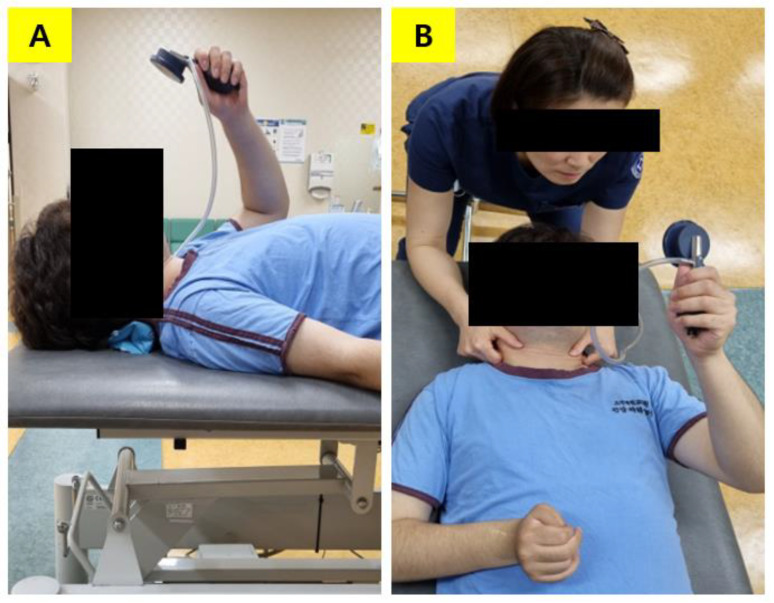
Contraction of deep neck flexor muscle progression. (**A**) The patient’s head and neck position using a pressure biofeedback device. (**B**) Bilateral contraction in supine.

**Figure 2 medicina-57-01312-f002:**
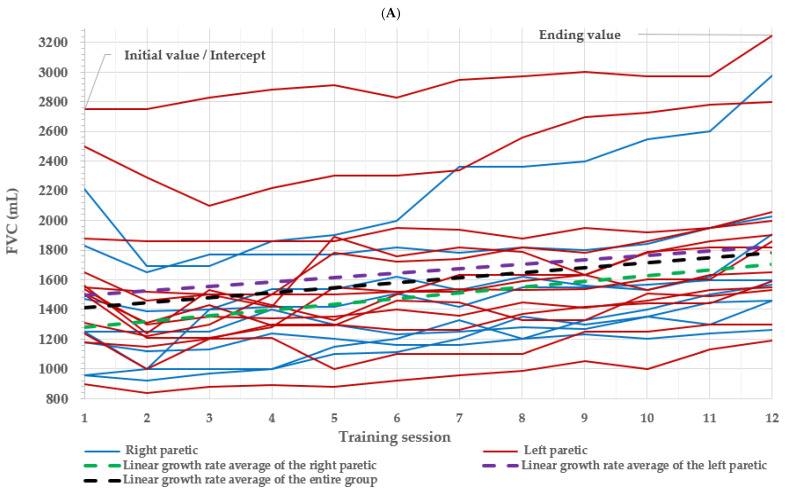
Initial value and ending value based on paralyzed side of hemiparetic stroke patients. Each solid line represents a respiration index value evolvement for one subject over 12 training sessions. The green dashed line represents the linear growth rate average of the right-sided paralysis group, the purple dashed line represents the linear growth rate average of the left-sided paralysis group, and the black dashed line represents the linear growth rate average of all stroke patients. (**A**) FVC, forced vital capacity; (**B**) FEV1, forced expiratory volume in the first second; (**C**) FEV1/FVC ratio, forced expiration ratio; (**D**) PCF, peak cough flow.

**Table 1 medicina-57-01312-t001:** Attachment site of deep neck flexor muscle.

Deep Neck Flexor Muscle		
Rectus capitis anterior	Origin	Anterior surface of the lateral mass of the atlas (C1 vertebra) and the root of its transverse process
	Insertion	The inferior surface of the occipital bone anterior to the foramen magnum
	Action	Aids in flexion of the head and the neck
Rectus capitis lateralis	Origin	Superior surfaces of the transverse processes of the atlas
	Insertion	Inferior surface of the jugular process of the occipital bone
	Action	Stabilizes the head; Weakly assists with lateral flexion of the head
Longus capitis	Origin	Anterior tubercles of transverse processes of C3–C6 vertebrae
	Insertion	Inferior surfaces of the basilar portion of the occipital bone.
	Action	Acting bilaterally: flexion of the cervical vertebrae and head; Acting unilaterally: rotation and lateral flexion of the cervical vertebrae and head to the same side
Longus colli	Origin	Upper fibers: anterior tubercles of the transverse processes of C3–C5; Central fibers: anterior surface of vertebral bodies of C5–T3 vertebrae; Lower fibers: anterior surface of vertebral bodies of T1–T3 vertebrae
	Insertion	Upper fibers: anterior tubercle of C1 (atlas); Central fibers: anterior surface of vertebral bodies of C2–C4 vertebrae; Lower fibers: anterior tubercles of the transverse processes of C5 and C6
	Action	Bilaterally, longus capitis acts as a weak flexor of the head and cervical vertebrae alone; unilateral action of the longus capitis muscle serves to rotate and tilt the cervical vertebrae and head to the ipsilateral side

**Table 2 medicina-57-01312-t002:** Composition of the null model.

Variable	Model	Formulas in the Model
FVC	Level-1 model	*FVC_mj_* = *ψ_0j_* + *ψ_1j_* × *(TIME_mj_)* + *𝑒_mj_*
	Level-2 model	*ψ_0j_* = *γ_00_* + *𝑢_0j_* *ψ_1j_* = *γ_10_* + *𝑢_1j_*
FEV1	Level-1 model	*FEV1_mj_* = *ψ_0j_* + *ψ_1j_* × *(TIME_mj_)* + *𝑒_mj_*
	Level-2 model	*ψ_0j_* = *γ_00_* + *𝑢_0j_* *ψ_1j_* = *γ_10_* + *𝑢_1j_*
FEV1/FVC	Level-1 model	*FEV1/FVC_mj_* = *ψ_0j_* + *ψ_1j_* × *(TIME_mj_)* + *𝑒_mj_*
	Level-2 model	*ψ_0j_* = *γ_00_* + *𝑢_0j_* *ψ_1j_* = *γ_10_* + *𝑢_1j_*
PCF	Level-1 model	*PCF_mj_* = *ψ_0j_* + *ψ_1j_* × *(TIME_mj_)* + *𝑒_mj_*
	Level-2 model	*ψ_0j_* = *γ_00_* + *𝑢_0j_* *ψ_1j_* = *γ_10_* + *𝑢_1j_*

FVC, forced vital capacity; FEV1, forced expiratory volume in the first second; FEV1/FVC, forced expiration ratio; PCF, peak cough flow.

**Table 3 medicina-57-01312-t003:** Composition of the research model.

Variable	Model	Formulas in the Model
FVC	Level-1 model	*FVC_mj_* = *ψ_0j_* + *ψ_1j_* × *(TIME_mj_)* + *𝑒_mj_*
	Level-2 model	*ψ_0j_* = *γ_00_* + *γ_01_* × *(PARETIC_j_)* + *𝑢_0j_* *ψ_1j_* = *γ_10_* + *γ_11_* × *(PARETIC_j_)* + *𝑢_1j_*
FEV1	Level-1 model	*FEV1_mj_* = *ψ_0j_* + *ψ_1j_* × *(TIME_mj_)* + *𝑒_mj_*
	Level-2 model	*ψ_0j_* = *γ_00_* + *γ_01_* × *(PARETIC_j_)* + *𝑢_0j_* *ψ_1j_* = *γ_10_* + *γ_11_* × *(PARETIC_j_)* + *𝑢_1j_*
FEV1/FVC	Level-1 model	*FEV1/FVC_mj_* = *ψ_0j_* + *ψ_1j_* × *(TIME_mj_)* + *𝑒_mj_*
	Level-2 model	*ψ_0j_* = *γ_00_* + *γ_01_* × *(PARETIC_j_)* + *𝑢_0j_* *ψ_1j_* = *γ_10_* + *γ_11_* × *(PARETIC_j_)* + *𝑢_1j_*
PCF	Level-1 model	*PCF_mj_* = *ψ_0j_* + *ψ_1j_* × *(TIME_mj_)* + *𝑒_mj_*
	Level-2 model	*ψ_0j_* = *γ_00_* + *γ_01_* × *(PARETIC_j_)* + *𝑢_0j_* *ψ_1j_* = *γ_10_* + *γ_11_* × *(PARETIC_j_)* + *𝑢_1j_*

FVC, forced vital capacity; FEV1, forced expiratory volume in the first second; FEV1/FVC, forced expiration ratio; PCF, peak cough flow.

**Table 4 medicina-57-01312-t004:** General characteristics of study subjects.

	Total (*n* = 21)	Rt. Side (*n* = 8)	Lt. Side (*n* = 13)	t	*p*
Sex (M/F)	4/17	1/7	3/10		
Age (years old)	76.00 ± 10.19	78.13 ± 9.08	74.69 ± 10.97	0.741	0.468
Height (cm)	159.86 ± 6.93	161.73 ± 4.95	158.72 ± 7.88	0.964	0.347
Weight (kg)	53.56 ± 9.27	49.85 ± 5.40	55.84 ± 10.55	−1.714	0.103
BMI (kg/m^2^)	20.96 ± 3.45	19.18 ± 2.91	22.05 ± 3.39	−1.987	0.061

Mean ± SD, mean ± standard deviation; M/F, male/female; BMI, body mass index; Rt., right; Lt., left.

**Table 5 medicina-57-01312-t005:** Descriptive statistics results of variable values.

		Mean (Standard Deviation, SD) Minimum to Maximum											
		1	2	3	4	5	6	7	8	9	10	11	12
Rt. Side	FVC	1388.75 (435.15) 960~2210	1252.50 (298.56) 920~1690	1326.25 (297.08) 970~1770	1403.75 (319.37) 1000~1860	1422.50 (294.07) 1100~1900	1456.25 (334.28) 1110~2000	1503.75 (400.39) 1160~2360	1547.50 (394.92) 1200~2360	1555.00 (392.17) 1230~2400	1598.75 (429.43) 1200~2550	1657.50 (438.92) 1240~2600	1783.75 (543.93) 1260~2980
	FEV1	1105.00 (263.60) 800~1620	1138.75 (276.79) 780~1490	1201.25 (301.64) 720~72	1222.50 (312.81) 740~1600	1278.75 (269.73) 830~1630	1271.25 (306.61) 820~1700	1345.00 (378.04) 840~2040	1370.00 (351.73) 1050~2040	1390.00 (347.85) 980~2020	1433.75 (406.59) 1000~2270	1445.00 (389.73) 980~2210	1522.50 (460.74) 1040~2450
	FEV1/FVC	80.88 (10.38) 56~88	90.88 (6.94) 78~98	90.13 (8.03) 72~97	86.75 (8.00) 74~99	89.88 (7.83) 75~99	87.13 (7.06) 74~98	89.13 (8.82) 70~99	89.00 (5.98) 80~98	89.25 (7.48) 75~98	89.50 (7.33) 75~98	87.13 (8.17) 75~98	85.50 (9.75) 71~97
	PCF	115.00 (62.96) 60~230	118.13 (59.46) 55~220	124.38 (59.67) 60~220	143.13 (68.76) 60~245	158.13 (78.28) 60~260	165.63 (80.38) 65~275	181.25 (84.12) 70~285	188.75 (82.71) 70~290	193.75 (79.36) 80~290	200.63 (79.93) 75~300	209.38 (81.96) 80~310	211.25 (84.97) 80~315
Lt. Side	FVC	1620.77 (509.39) 900~2750	1473.08 (534.09) 840~2750	1530.00 (496.96) 880~2830	1547.69 (512.22) 890~2880	1610.77 (547.81) 880~2910	1633.85 (505.05) 920~2830	1662.31 (529.89) 960~2950	1693.08 (549.63) 990~2970	1716.92 (556.77) 1050~3000	1756.92 (548.70) 1000~2970	1803.85 (535.76) 1130~2970	1884.62 (574.10) 1190~3250
	FEV1	1350.77 (502.92) 650~2460	1334.62 (491.00) 710~2420	1353.08 (482.67) 700~2620	1396.15 (508.32) 700~2720	1457.69 (547.31) 700~2740	1476.15 (472.08) 740~2520	1468.46 (499.73) 830~2750	1528.46 (520.08) 800~2670	1562.31 (516.17) 950~2700	1513.15 (619.40) 810~2570	1578.62 (617.86) 920~2570	1692.31 (545.85) 950~2810
	FEV1/FVC	82.00 (11.92) 63~97	90.38 (7.22) 77~99	87.85 (8.02) 75~99	88.54 (6.86) 76~97	89.31 (6.13) 78~98	89.77 (5.67) 80~98	88.38 (6.68) 74~98	90.69 (6.92) 74~98	90.77 (5.95) 76~98	89.08 (6.22) 80~98	91.08 (5.92) 70~290	89.31 (6.47) 79~98
	PCF	112.69 (50.23) 60~220	108.08 (43.04) 50~180	115.77 (54.23) 55~240	126.54 (56.91) 60~250	131.92 (59.78) 65~250	145.77 (63.60) 70~250	153.46 (62.96) 70~250	166.92 (62.10) 70~250	176.92 (62.70) 70~270	184.23 (71.58) 70~280	189.62 (73.58) 70~290	193.85 (75.34) 70~300

FVC, forced vital capacity; FEV1, forced expiratory volume in the first second; FEV1/FVC, forced expiration ratio; PCF, peak cough flow.

**Table 6 medicina-57-01312-t006:** Changes in respiratory function exhibited by the entire group.

Variable	Fixed Effect	Coefficient	Standard Error	t-Ratio
FVC	Initial value average (*ψ_0j_*)	1413.071	94.740	14.915 **
	Linear growth rate average (*ψ_1j_*)	33.345	4.834	6.898 **
	Random Effect	Variance Component	Standard Deviation	$x^{2}$
	Within the group (*𝑢_0_*)	195,160.550	441.770	1437.965 **
	Intergroup (*𝑢_1_*)	449.946	21.212	157.849 **
	Error variance (*𝑒*)	9335.194	96.619	
Variable	Fixed Effect	Coefficient	Standard Error	t-Ratio
FEV1	Initial value average (*ψ_0j_*)	1241.894	88.829	13.981 **
	Linear growth rate average (*ψ_1j_*)	31.086	5.823	5.339 **
	Random Effect	Variance Component	Standard Deviation	$x^{2}$
	Within the group (*𝑢_0_*)	170,456.050	412.863	985.404 **
	Intergroup (*𝑢_1_*)	663.913	25.767	178.554 **
	Error variance (*𝑒*)	11,975.679	109.433	
Variable	Fixed Effect	Coefficient	Standard Error	t-Ratio
FEV1/FVC	Initial value average (*ψ_0j_*)	87.068	1.514	57.494 **
	Linear growth rate average (*ψ_1j_*)	0.269	0.127	2.114 *
	Random Effect	Variance Component	Standard Deviation	$x^{2}$
	Within the group (*𝑢_0_*)	43.581	6.602	144.742 **
	Intergroup (*𝑢_1_*)	0.191	0.438	43.103 **
	Error variance (*𝑒*)	23.696	4.868	
Variable	Fixed Effect	Coefficient	Standard Error	t-Ratio
PCF	Initial value average (*ψ_0j_*)	107.210	11.612	9.233 **
	Linear growth rate average (*ψ_1j_*)	9.043	1.091	8.291 **
	Random Effect	Variance Component	Standard Deviation	$x^{2}$
	Within the group (*𝑢_0_*)	2925.032	54.084	1234.095 **
	Intergroup (*𝑢_1_*)	25.085	5.008	459.039 **
	Error variance (*𝑒*)	163.409	12.783	

* *p* < 0.05, ** *p* < 0.01; initial value average (for intercept 1, *ψ_0_* intercept 2, *γ_00_*); linear growth rate average (for *TIME* slope, *ψ_1_* intercept 2, *γ_10_*); within the group (intercept 1, *𝑢_0_*); intergroup (*TIME* slope, *𝑢_1_*); error variance (level-1, *𝑒*); FVC, forced vital capacity; FEV1, forced expiratory volume in the first second; FEV1/FVC ratio, forced expiration ratio; PCF, peak cough flow.

**Table 7 medicina-57-01312-t007:** Changes in respiratory function based on paralyzed side of stroke patients.

Variable	Fixed Effect	Side	Coefficient	Standard Error	t-Ratio
FVC	Initial value average (*ψ_0j_*)	Rt.	1278.349	103.405	12.363 **
		Lt.	217.627	169.343	1.285
	Linear growth rate average (*ψ_1j_*)	Rt.	38.728	9.620	4.026 **
		Lt.	−8.696	10.782	−0.806
	Random Effect	Variance Component		Standard Deviation	$x^{2}$
	Within the group (*𝑢_0_*)	193,232.184		439.582	1352.756 **
	Intergroup (*𝑢_1_*)	457.354		21.386	152.112 **
	Error variance (*𝑒*)	9335.194		96.619	
Variable	Fixed Effect	Side	Coefficient	Standard Error	t-Ratio
FEV1	Initial value average (*ψ_0j_*)	Rt.	1114.904	89.207	12.498 **
		Lt.	205.138	155.889	1.316
	Linear growth rate average (*ψ_1j_*)	Rt.	35.529	11.040	3.218 **
		Lt.	−7.176	12.755	−0.563
	Random Effect	Variance Component		Standard Deviation	$x^{2}$
	Within the group (*𝑢_0_*)	168,644.686		410.664	926.388 **
	Intergroup (*𝑢_1_*)	689.840		26.265	175.508 **
	Error variance (*𝑒*)	11,975.679		109.433	
Variable	Fixed Effect	Side	Coefficient	Standard Error	t-Ratio
FEV1/FVC	Initial value average (*ψ_0j_*)	Rt.	87.454	1.929	45.342 **
		Lt.	−0.623	2.878	−0.217
	Linear growth rate average (*ψ_1j_*)	Rt.	0.086	0.209	0.411
		Lt.	0.296	0.259	1.142
	Random Effect	Variance Component		Standard Deviation	$x^{2}$
	Within the group (*𝑢_0_*)	46.141		6.793	144.467 **
	Intergroup (*𝑢_1_*)	0.187		0.433	40.493 **
	Error variance (*𝑒*)	23.696		4.868	
Variable	Fixed Effect	Side	Coefficient	Standard Error	t-Ratio
PCF	Initial value average (*ψ_0j_*)	Rt.	114.095	20.600	5.539 **
		Lt.	−11.121	24.736	−0.450
	Linear growth rate average (*ψ_1j_*)	Rt.	9.701	1.508	6.431 **
		Lt.	−1.063	2.118	−0.502
	Random Effect	Variance Component		Standard Deviation	$x^{2}$
	Within the group (*𝑢_0_*)	3049.280		55.220	1221.383 **
	Intergroup (*𝑢_1_*)	26.171		5.116	454.143 **
	Error variance (*𝑒*)	163.409		12.783	

** *p* < 0.01; initial value average (for intercept 1, *ψ_0_*): Rt. (intercept 1, *𝑢_0_*), Lt. (*PARETIC*, *γ_01_*); linear growth rate average (for *TIME* slope, *ψ_1_*): Rt. (intercept 2, *γ_10_*), Lt. (*PARETIC*, *γ_11_*); within the group (intercept 1, *𝑢_0_*); intergroup (*TIME* slope, *𝑢_1_*); error variance (level-1, *𝑒*); FVC, forced vital capacity; FEV1, forced expiratory volume in the first second; FEV1/FVC ratio, forced expiration ratio; PCF, peak cough flow.

## Data Availability

All data relevant to the study are included in the article. Data were collected from studies published online or publicly available, and specific details related to the data will be made available upon request.

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
