# Peer review of "Neck Stabilization Exercises Enhance Respiratory Function after Stroke: Respiratory Function Index Change Trajectory Analyzed Using a Hierarchical Linear Model"

_medicina, 2021, doi:10.3390/medicina57121312_

Round 1

Reviewer 1 Report

The authors have addressed all my comments and I am satisfied with the revision. Below are two additional comments for consideration

  1. There are 4 subplots in figure 2(A,B,C,D). When referring to figure 2 in the main text, please use figure 2A, 2B etc, instead of using figure. 2 alone
  2. I notice that a photo that included a patient is added as figure 1. Although the patients' eyes have been covered, I think the patient can still be identified by people who know him. Please cover the whole face of the patient. 

Author Response

We attached the file.

Reviewer 2 Report

The authors did a great job addressing most of my comments. The clarity of the article improved significantly. 

There are two places that could still use some improvements:

Comment R01 - figure 2 clarity

  • The author should explain in text or figure capture what each element in the figure represents (e.g., each solid line represents a respiration index value evolvement for one subject over 12 training sessions).
  • The last dot (at session 12) should be ‘ending value’, not ‘Amount of change’. The ‘amount of change’ should be ‘ending value – initial value’.
  • Figure legend is still not clear. Use ‘right paretic’ instead of ‘right’.
  • The dashed line should be ‘linear regression fitted line/linear growth rate/slope’ for 'one representative subject'. Not for all points.

Comment R02 - Variable names/font

When describing the hierarchical linear model, a lot of variables (such as ψ0j, TIMEmj, PARETICj, etc. ) were used in both equations in Table 2 & Table 3 and in text, and the fonts and style used were not consistent. For all the variables, consider following the commonly used math font standard:

  1. Use MathType or math equation function to input variable names/equations. Use a common math font, e.g., Cambria Math.
  2. Make sure to be consistent on italic/regular
  3. For variables like 'TIMEmj', 'PARETICj', use a proper subscript expression instead of using uppercase/lowercase.

Author Response

We attached the file.

This manuscript is a resubmission of an earlier submission. The following is a list of the peer review reports and author responses from that submission.

Round 1

Reviewer 1 Report

Thank you for inviting me to review this manuscript. This study aimed to investigate the effects of neck stabilization exercise on people with stroke and the influence of the paretic side on the growth rate. Results of the study showed that a significant difference was found in the respiratory function analysis growth rate of the entire group (p<0.05). With the side of paralysis factored into the analysis, the results show that patients with right-side paralysis demonstrated a greater improvement in lung functions. This manuscript provides evidence to support that neck stabilization training is longitudinally effective in improving respiratory function in stroke patients with right-side paralysis. The training was more effective in stroke patients with right-sided weakness than left-sided weakness.

In general, the manuscript is easy to understand. The research question is interesting, and only a few similar studies have been done in this area. It seems that the study had been conducted properly.

Please find my specific comments below:

Introduction

The authors tried to explain the mechanisms on how the imbalance in neck stabilization leads to poor pulmonary function and respiratory function. May I know if this approach has been studied in stroke or other populations? And what are the results?

I think the description of the HLM can be moved to ‘Statistics’ or the discussion session.

One of the focuses of this manuscript is the effect of the ‘paretic side’ on neck stabilization training. The author should propose why this factor is important and should be explored in this manuscript.

Method

The author did not mention how did they educate the subjects to limit the use of accessory muscle during training.

The first paragram of section 3.1.1. Null model and first two paragrams of section 3.1.2. The research model, which descript the statistical model,  should be moved to the ‘Methods’ session

Results

Line graphs showing the actual changes of lung function at each time points along the whole study period helps the reader to understand the effects of the training.

There is two subplots in figures 1 – 4, illustrating the whole model and the model with the ‘side of paralysis’ as an additional factor. I would suggest removing subplot A since it does not provide any additional information.

Discussion

Please comment on the patient’s lung function, is the study cohort having a good or bad lung function?

Author Response

We attached the file.

Reviewer 2 Report

Summary

The author did an interesting research study to investigate the effect of longitudinal neck stabilization exercise on the stroke patient’s respiratory functions. By applying a two-level hierarchical linear model (that can account for inter-group/subject individual difference) that factored in the side of paralysis information in data analysis/statistics, the author found that the 12-session neck stabilization exercise could help stroke patients improve their respiratory function, and the improvement is more significant in the stroke patients with right-side paralysis. Despite the small sample size, this interesting finding possibly revealed the differences/imbalance of two sides in respiratory function-related muscle decrease and could potentially inform the future exercise design to maximize the therapeutic benefits.

Overall, the paper is well-structured scientifically and the story is clearly told. However, some core concept/equation/method used is not well-explained/lacking, which make the paper quite hard to follow. The figures and tables are hard to read. Although the facts were presented, the key information was not emphasized clearly. Mixed-use of terms, variable names are also quite hard to follow. I had a hard time understanding the model equations, table rows, and column labels the first time I read the manuscript and had to jump between different sections to try to understand the terms. In short, it feels like although the author understands what he/she does and structured the article well, he/she did not put enough thoughts when drafting the paper to accommodate the general readers.

While I do think this interesting work has research value worth sharing, it needs a lot of revisions to improve the clarity and readability. Please see my comments down below.

P.S.: gray, italic font (e.g., “Neck stabilization exercise”) indicates a direct citation from the manuscript. 

Detailed Comments

Comment 01 – Page 1 - Paper title might be misleading

The title of the manuscript is “Neck stabilization exercise for respiratory function after stroke: trajectory analysis using hierarchical linear model”.

Without the detailed context or explanation, the term “trajectory” that appeared here in the title for the first time could be misleading. After reading the whole article, I know the author means “respiratory function measurement index score growth/change trajectory”, but when you read this for the first time without the context, you may think the ‘trajectory’ as position/coordinates in 3D space, e.g., neck trajectory (neck position coordinates (position x y z) in 3D space).

I think it would be more clear if either you explicitly say it is “respiratory function index change trajectory”, or, replace the term “trajectory” with “trend” (or similar words) to avoid confusion.

Comment 02 – Hierarchical Linear Model needs to be better explained.

Hierarchical Linear Model (HLM) is the core analysis of this research paper and understanding the method is crucial to understanding the paper and findings. Therefore, a simple yet detailed introduction of the method (advantage, feature, and model formula) is essential.

Aside from citing two papers ([17] & [18]) that introduce the HLM, the author did briefly introduce the reason and advantage of using the HLM, and its mathematic model/equation. However, the model equation and coefficients in the formula were not presented detailed enough in the text and in Table 4 & Table 6 row indexes, which makes the paper hard to follow sometimes.

Here are a few places that could use some improvements:

Page 4, 3.1.1 Null model, paragraph 2

“It also determines whether a model that explains differences between individuals by inputting independent variables to Level-2 of the basic model in the future research model is meaningful [22].”

This sentence is very long and hard to follow. Do you mean:

  • “It also determines whether a model is meaningful”
  • “This model explains differences between individuals”
  • “It achieves that by inputting independent variables (Time) to Level-2 of the basic model in the future research model”

Please consider using multiple short sentences to increase clarity and avoid confusion.

Page 5, line 1, 3.1.1 Null formula

“ψ1j is the coefficient of the influence of Xij with the level-1 coefficients.”

Xijis not defined and does not appear in any formula. What is it? It is the independent variable (TIMEmj)? Please be consistent and do not use variable names that do not appear in the context.

Page 5, line 3 ~ 4, 3.1.1 Null formula

“The 2-level model is used to predict or explain the one-level coefficients ψ0j and ψ1j.”

The expression (“2-level” and “one-level”) does not feel right. Do you mean “The level-2 model is used to predict or explain the level-1 coefficients ψ0j and ψ1j.”

Page 4 ~ 5,  3.1.1 Null model,  paragraph 2

The null model equation/formula was not detailed explained.

“… and ψ0j is a regression coefficient for the influence of the mean with level-1 coefficients. ψ1j is the coefficient of the influence of Xij with the level-1 coefficients.”

  • Is “ψ0j the intercept/initial value?
  • Is “ψ1j the fitted line’s slope rate/“(linear) growth rate”? If not, what is the “(linear) growth rate”?

These terms (initial value, growth rate, etc.) were used frequently in the text but are not explicitly defined. It is important to explain clearly what they represent and it would be great if the author could specify which coefficient/variable those terms correspond to in the regression model equation/formula.

Also, the author explained all the coefficients in the Level-1 model in Table 3 in the second paragraph in the section “3.1.1 Null model”. However, the coefficients of the Level-2 model (γ00 and u0j, γ10 and u1j) were not explained in the text, which makes the model concept hard to follow for readers who are not familiar with the HLM.

Page 5, paragraph 2

“The linear growth rate of the FVC score of all stroke patients during the measurement period was 33.345, which was statistically significant.”

When is a “linear growth rate” considered significant? Did you use the t-ratio values to determine if the growth rate is significant? If so, what is the threshold?

Comment 03 – Page 1, last 4 lines:

“including forced vital capacity (FVC), forced expiratory volume in the first second (FEV1), forced expiration ratio (FEV1/FVC), and peak cough flow (PCF),”

As these four measurements are the main respiratory function assessment indexes used in this study, it would be great to briefly introduce what they represent/how they reflect a certain aspect of respiratory function when they first appear in the article (page 1, last 4 lines) instead of explaining them later (page 3, section 2.3 Measurement). Or, add parenthesis and state where you can find more detailed explanation of those terms, such as: “including forced vital capacity (FVC), forced expiratory volume in the first second (FEV1), forced expiration ratio (FEV1/FVC), and peak cough flow (PCF) (please refer to Section 2.3 Measurement for a more detailed explanation.)

Comment 04 – Table 4 & Table 6 needs to be improved

The row and column index labels in Table 4 and Table 6 are “initial value average”, “linear growth rate average”, “Within the group variant component”, “Intergroup variant component” and “Error variance component”. While these terms show what effect we are checking, it is easy to get lost which coefficient/variable they correspond to in the model formula (in Table 3 and Table 5).

Please consider adding the corresponding coefficient and variable symbol/letter (ψ0j, ψ1j, emj, γ00, u0j, γ10, u1j) from the model formula to the Table 4 and Table 6 row/column index names, for example:

  • “Initial value average (ψ0j)”
  • “Linear growth rate average (ψ1j) “

Also, in Table 4’s footnote:

“Initial value average (for INTRCPT1, ψ0     INTRCPT2, γ00)”

The “INTRCPT1” “INTRCPT2” is not defined. What do they represent? Is it short for “intercept 1” and “intercept 2”? On page 4, the last paragraph, when describing the null model, the author states the “ψ0j is a regression coefficient”, but here, it is referred to as an “intercept”. I could guess what the author was referring to but it would be great if you could keep it consistent across the article to avoid confusion.

Comment 05 – Page 2, last paragraph

“Therefore, unlike previous studies, this study aimed to observe the degree and effect of longitudinal changes in respiratory function at the group and individual levels by applying neck stabilization training to stroke patients.”

This long sentence is very confusing and hard to follow. Do you mean:

“aimed to observe the effects of applying neck stabilization training to stroke patients, check the degree of longitudinal changes in respiratory function, at both the group level and individual level (thanks to the hierarchical linear model analysis)?

Comment 06 – Page 3, 2.2 Training method.

This section briefly explains how neck stabilization training was performed (strengthing deep neck muscle by providing constant contractile force via a biofeedback device, and relaxing the shallow respiratory auxiliary muscles and cranial neck). As a general reader who is not very familiar with human anatomy, I would be curious about where are those muscles and it would be great if the author could include a figure here to demonstrate the location of those muscles, and which one is stimulated/strengthened.

Comment 07 – Page 4, line 1-2

“For the experimental effect analysis, a multi-layered growth model was implemented to check how the study variables at the paraplegic side level and the stroke patient level individually affect respiratory function index.”

This is a bit confusing. What are the study variables at the “paraplegic side level” and “stroke patient level” study variables?

Is the study variable at the “paraplegic side level” the independent variables at model level-2, i.e., “PARETICj”?

Are the study variables at the “stroke patient level” the regression model coefficients (intercept ψ0j, slope rate ψ1j, and error emj)?

Do you mean, you implemented a multi-layered growth model to investigate whether 1) which paretic leg side and 2) which individual may have a significant linear respiratory function index score growth over time? Please clarify and be clear.

Comment 08 – Page 4, Table 1

What is the statistical test used in Table 1 to compare the left- and right- paralyzed groups? Did you use a two-sample t-test and report the t and p values? If so, did you test the data normalcy before using the parametric test?

Aside from this, at multiple places throughout this article, inter-group significant differences and line fitting significances were reported many times. However, the statistical test method applied (e.g., two-sample t-test), and the criteria used to determine the significance (e.g., p < 0.05) are lacking. Please add that information to the section “2.4 Data analysis” or section “2.5 Statistical analysis” and mention again at the place in text when it is reported.

Comment 09 – Page 4, Table 2

Are those numbers calculated based on the entire subject group after the neck stabilization exercise session-1? If so, what do those numbers tell us? Were they presented just to give the reader an idea of the initial values/intercept levels of each respiratory function index for each patient?

Comment 10 – Page 4, 3.1.1 Null model

“It also determines whether a model that explains differences between individuals by inputting independent variables to Level-2 of the basic model in the future research model is meaningful [22]”

This sentence is way too long and hard to follow. I tried to guess what the author was trying to say and this is my understanding:

  1. “It also determines whether a model is meaningful”.
  2. “This model explains differences between individuals (i.e., left-paretic group v.s. right paretic group)”.
  3. “It (the second point) is achieved by inputting independent variables (PARETICj) to Level-2 of the basic model in the future research model”.

Is this correct? Please consider using multiple short, clear sentences to avoid confusion.

Also, when describing a methodology in text, instead of using only the general term, always try to use the “general term + specific variable names used in this study” combination to improve the clarity of the article.

For example, instead of using “general name”:

  • “Level-1 model independent variable”
  • “Linear growth rate”
  • “Level-2 independent variable”

Try to use “general name (corresponding variable in this study/article)” and be specific:

  • “Level-1 model independent variable (TIMEmj)”
  • “Linear growth rate (ψ1j)”
  • “Level-2 independent variable (PARETICj)”

Comment 11 – Page 6 - Possible typo

“PARETICj is a two-level independent variable and is a characteristic variable of group j.”

Is “PARETICj” an independent variable that’s going to affect the dependent variables at both two levels? If so, “two-level” is correct.

If you mean the “PARETICj” is an independent variable in the level-2, and only affects the dependent variable in the level-2, then you should rephrase the sentence as:

“PARETICj is a level-2 independent variable

Comment 12 – Page 6 – Possible inaccurate expression.

In “3.1.2, Research model” and the results description paragraphs:

“There was a significant difference seen in the initial FVC scores of the left- and right-sided paralysis groups. The growth rate of the former decreased negatively to -8.696 (p>0.05), and the growth rate of the latter increased positively to 38.728 (p<0.01).”

When stating the growth rate increased/decreased, what are you comparing the current value (growth rate value) to? What is the base number you are increased/decreased from? Is it the growth rate of the “whole group”?

Do you mean, compared to the “whole group FVC growth rate (33.345)”, the “left-paretic group FVC group growth rate (-8.696)” is smaller (even becomes negative), and the “right-paretic group FVC group growth rate (38.728)” is larger?

As these values are linear regression models coefficients fitted to the same group/subgroup of population at the same time, there is no “changes/improvements/increase/decrease” involved. “Inter-group difference” would be a more accurate description.

Considering changing the “increased positively” and “decreased negatively” to “is bigger than (whole group value)” and “is smaller than the whole group value (and becomes a negative rate (<0))”

Comment 13 - Table 4 and Table 6, term notation

In Table 4 and Table 6, the row labels are the names of the regression model coefficients (“Initial value average”, “Linear growth rate average”, “Within the group random effects”) and the corresponding variable names in the model formula is given in the table footnote annotation. This could be hard to follow for the reader who is not familiar with the HLM.

Considering putting both together in the table row label, such as:

  • “Initial value average (ψ0 for level-1; γ00 for level-2)”
  • “Linear growth rate average (ψ1 for level 1; γ10 for level-2)”

Another question is:

How did you determine whether the growth rate of a fitted model/line is significant or not? Did you achieve that by checking t-ratio values in Table 4 & Table 6? If so, what was the threshold used?

Comment 14 – Figure 1 – Figure 4

Figures are hard to read and need to be improved.

Axis ticks labels are too small to read. The x-axis and y-axis labels lack the units. The x-axis range from 0~11. Do they correspond to 12 neck stabilization training sessions? If so, the x-axis is the training session index, not the time (as you have 3 training/week for 4 weeks, the intervals between the sessions are not consistent). If it is the session index, they could only be integers ranging from 1~12, not 0~11. Therefore, the x-axis label should be “training session index”, and the x-axis tick should only occur at integer locations (i.e., there should not be ‘2.75’, ‘5.5’, ‘8.25’).

What does each line represent? Does each line represent one subject? Every subject participated in a 12-session neck stabilization training. For each respiratory function index (FVC, FEV1, FVC/FEV1 ratio, and PCF), there are 12 measured values (1 value post each training). Is each line a linear regression model fitted to those 12 values from one subject?

Figure legend is also hard to understand. Consider directly stating “line color-paretic side” relationship, i.e., use “blue = right-paretic; red = left-paretic” directly instead of using “blue: PARETIC = 0; red: PARETIC = 1” in the figure legend and then further stating “PARETIC = 0, Right side; PARETIC=1, Left side” in the figure description.

Also, to further improve the clarity and readability of the paper, it would be great to pick one line, and mark/annotate its initial value/intercept, and linear growth rate/line slope rate so the reader could easily know what exactly do those terms represent?

Comment 15 – Page 10 – Possible typo

In “4. Discussion”, paragraph 1:

“In this study, FVC, FEV1, FEV1/FVC ratio, and PCF were used to investigate the effect of neck stabilization exercise on group and individual respiratory function growth trajectory, also factoring inside of paralysis of stroke patients.”

I believe the author was trying to say “… also factoring in the side of paralysis  of stroke patients.” Instead of “… also factoring inside of paralysis  of stroke patients.”

Comment 16 – Page 10 – Possible inaccurate expression

In the “4. Discussion” section, paragraph 2:

“… the growth rate of respiratory function of patients with left-sided paralysis decreased negatively while the growth rate of the respiratory function of patients with right-sided paralysis showed increased positively.”

Assuming the (linear) growth rate of the respiration function is the slope rate of the fitted line (i.e., ψ1j in the formula FVCmj = ψ0j + ψ1j*(TIMEmj) + emj), you should say either:

  • “The slope/linear growth rate is positive/negative” as the “growth rate” already indicates it’s a score change over time.

Or

  • “The respiratory function index (i.e., FVC) score increase/decrease over time.”

You should NOT say the “the growth rate of … increase positively/decrease negatively.”, as this sounds like “the rate of the growth rate”, which is the 2nd order derivative of the respiratory function index score.

This statement also appeared on page 10, 3rd-to the last line. “it is judged that although the respiratory function was longitudinally effective, the growth rate gradually decreased.”

This makes it sound like the respiratory function score still keeps growing, only that its growing speed slowed down over time, but still grows. (i.e., the slope of the line slowly change from a big positive number to 0)

Comment 17 – page 11

“it was found that the repeated application of neck stabilization training was more effective for improving respiratory function in patients with right-sided paralysis than in those with left-sided paralysis.”

As the author mentioned, given the very small sample size and two not certain reasons speculated that might lead to this finding, it is probably a bit too bold to make such a strong statement (“neck stabilization training is more effective for improving respiratory function in patients with right-sided paralysis than those with left-sided paralysis”. Consider slightly toning down the statement.

Author Response

We attached the file.
